# Deaths from undetected COVID-19 infections as a fraction of COVID-19 deaths can be used for early detection of an upcoming epidemic wave

**Yashaswini Mandayam Rangayyan**[1]*, **Sriram Kidambi**[2], **Mohan Raghavan**[1]

**1** Department of Biomedical Engineering, Indian Institute of Technology - Hyderabad, Hyderabad, Telangana, India, **2** Department of Natural Sciences and Mathematics, The University of Texas at Dallas, Richardson, Texas, United States of America

* bm20resch02003@iith.ac.in

## Abstract

With countries across the world facing repeated epidemic waves, it becomes critical to monitor, mitigate and prevent subsequent waves. Common indicators like active case numbers may not be sensitive enough in the presence of systemic inefficiencies like insufficient testing or contact tracing. Test positivity rates are sensitive to testing strategies and cannot estimate the extent of undetected cases. Reproductive numbers estimated from logarithms of new incidences are inaccurate in dynamic scenarios and not sensitive enough to capture changes in efficiencies. Systemic fatigue results in lower testing, inefficient tracing and quarantining thereby precipitating the onset of the epidemic wave. We propose a novel indicator for detecting the slippage of test-trace efficiency based on the number of deaths/hospitalizations resulting from known and hitherto unknown infections. This can also be used to forecast an epidemic wave that is advanced or exacerbated due to a drop in efficiency in situations where the testing has come down drastically and contact tracing is virtually nil as is prevalent currently. Using a modified SEIRD epidemic simulator we show that (i) Ratio of deaths/hospitalizations from an undetected infection to total deaths converges to a measure of systemic test-trace inefficiency. (ii) This index forecasts the slippage in efficiency earlier than other known metrics. (iii) Mitigation triggered by this index helps reduce peak active caseload and eventual deaths. Deaths/hospitalizations accurately track the systemic inefficiencies and detect latent cases. Based on these results we make a strong case that administrations use this metric in the ensemble of indicators. Further, hospitals may need to be mandated to distinctly register deaths/hospitalizations due to previously undetected infections. Thus the proposed metric is an ideal indicator of an epidemic wave that poses the least socio-economic cost while keeping the surveillance robust during periods of pandemic fatigue.

**Data Availability Statement:** The codes used in the manuscript are available at the following link as a public Git repository: https://gitlab.com/yashaswinimr1/trackarona.git.

**Funding:** The author(s) received no specific funding for this work.

**Competing interests:** The authors have declared that no competing interests exist.

## Introduction

India witnessed a huge part of the population affected by the COVID-19 virus in 3 waves from March 2020 to March 2022 [1] leading to a major crisis in saving lives, handling healthcare resources as well as suffering economic losses. Several administrative measures like lockdowns, social distancing, quarantining etc were implemented to curb the spread of these infections. These strategies should be invoked appropriately based on insights from epidemiological models. In the recent outbreak of COVID-19 parameters like the reproductive number (R0), serial interval, incubation period, transmissible period, proportion of confirmed cases in the population, proportion of deaths in critical cases and beta are some salient parameters used for modeling [2–4]. While these are some epidemiological parameters, COVID-19 is also characterized by many biological parameters like variance in transmissibility period and immunological strength across different age groups and viral strains, effect of vaccination etc [4, 5]. But these parameters are hard to study in an ongoing epidemic [5]. The authors highlight the complications of fitting the model to reported cases, as the cases can be distributed unevenly geographically. This would mean that there would be a sizeable population having latent infections that escapes being reported. Hence the authors suggest that hospitalizations and deaths are more reliable [5]. This way deaths and hospitalizations that get diagnosed with the infection are from the pool of latent infections thus acting as an indicator of latent infections.

The reasons for undetected cases are many. People who are asymptomatic or with limited access to healthcare (forming a large part of the population) might go undetected easily and hence the number of confirmed, recovered and deceased cases that are reported everyday can be an underestimate of the true figures [6–8]. Further insufficient testing, contact tracing or quarantining may contribute to this inefficiency. Very few techniques like PCR and antibody tests are used to estimate the number undetected infections in a particular population [9]. These techniques are highly randomized and can only estimate the undetected cases at specific time point making the data very sparsely sampled [9].

Metrics like number of Active cases and deaths are looked at for studying the case growth patterns in different Indian states and ranking them based on severity of the spread [10]. A good amount of time has elapsed before some insights can be drawn from these metrics. [11] describes the correlation between reproductive rate and test positivity rates showing how testing failed to catch up with the reproductive rate. Further contact tracing is also biased to the serial interval in symptomatic people [4] Despite being capable of predicting an impending wave, these metrics are sensitive to testing strategies and cooperation by the population in different states making them unreliable.

Amidst all the chaos of asymptotic cases, under-reporting etc, death is an unmissable event in an epidemic. The number of deaths in an epidemic provides a realistic picture of the extent of infection still latent in the population [5, 12, 13]. While it is intuitively recognized that detection of infection postmortem or at first presentation at intensive care units are an indicator of the gravity of the situation, this has never been quantified or studied as a measure or indicator of relevant quantities. Current reporting practices do not capture 'infections detected postmortem' or 'previously undetected infections presenting at ICU' separately from the detected infections. We propose that these metrics are vital indications of the health of a community or society and can be an early indicator of an emerging epidemic wave.

Modified SEIRD model [14] describes most of the above parameters in its model and uses the epidemic simulator as a tool for administration and governance. It uses compartments in two parallel tracks representing the quarantined and unquarantined population which makes it relevant in the days of COVID-19 pandemic. This model demonstrated the utility of using simulators for predicting the number of infections in both the quarantined and unquarantined

arms of the model in the normal course and in response to a variety of administrative actions, thereby estimating the effectiveness of administration reforms. These models empower the policy makers and health workers to understand the model parameters better and take necessary and timely interventions. It is well known that testing helps flatten the curve while undetected and latent cases spur on a full blown epidemic. The epidemic simulations [14] helped to quantify this effect as a measure of inefficiency.

Using the epidemic simulator described in [14], we demonstrate the utility of such metrics in early detection of an epidemic wave that is hastened or exacerbated by inefficiency [14] on account of insufficient testing, tracing or containment measures. These metrics that ensure robust surveillance of the pandemic and yet have a low cost of compliance are especially advantageous in scenarios of pandemic fatigue that is prevalent currently. We use the symmetry in the two arms of the model [14] along with the fact that death of either kind is unmissable, to estimate the inefficiency (or latent spread) of an epidemic. We show here that recording the number of 'postmortem detection of infection' as a proportion of total deaths can be a good metric. The ratio of 'previously undetected infections at hospitalization' to total hospitalizations would also serve a similar purpose. We hereby make a case for modifying the Standard Operating Procedures to make this book keeping possible.

## Methods

Compartmental epidemiology models like SIS, SIR, SEIR and SEIRD model the effects of changing dynamics in an epidemic in the form of transitions through a sequence of compartments [15–18]. These transitions are a result of changes in number of individuals in each state as the individual subjects move from susceptible to getting exposed, infected and finally recover or die. The reproductive number R0 in an epidemic is the mean number of infections an infected person can cause in a fully susceptible population during the lifetime of a person's infection [19]. R0 indicates the attack rate or spread of infection with values greater than 1 indicating build-up and values less than 1 indicating fade out of the disease [19–21]. This metric is useful in gauging the risk of an epidemic [2–4, 22] and when estimated in real time can determine the effectiveness of intervention strategies like lock downs and isolation that change the transmission dynamics [20, 22].

The Modified SEIRD model described in [14] uses two symmetric, parallel arms to represent the quarantined and free populations. The transition rates from one compartment to another are proportional to the extent of contact tracing and self-reporting in each of the respective compartments. These transition rates are described in Fig 1 [14].

These transitions provide a metric called the intervention inefficiency that is interpreted as the ineffectiveness of interventions; or the fraction of infections that are undetected. The inefficiency being a fraction in [0, 1] is the complement of efficiency [14].

$$Intervention\ Inefficiency = (1-c)\{a(1-c)+(1-a)(1-q)\} \tag{1}$$

where,

a -The fraction of infection spread through contacts,
q -Fraction of infections detected through random testing and self reporting,
c -Fraction of infections detected through contact tracing.
It is observed that c and q play a significant role in altering the inefficiency.

The ratio of unreported cases to total number of cases can be captured in any of the compartments of the model at I or beyond, to measure the inefficiency. Since death is an unmissable event in an epidemic, the data for it is more easily and accurately available than in any other compartment. Owing to the limitations in estimating the number of undetected cases in

## Modified SEIRD model

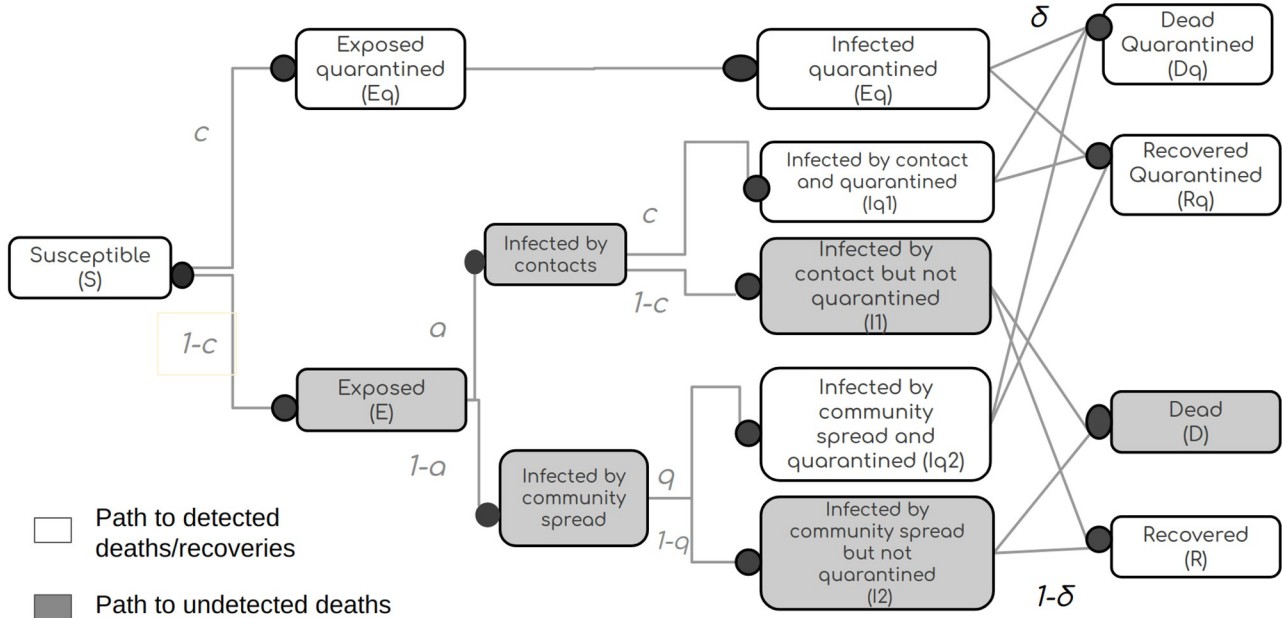

**Fig 1. Figure adapted from [14] describes the fraction of population flowing through each compartment in the parallel arms of the modified SEIRD model.**

the population as seen in the introduction, we use the model simulations to obtain the number of undetected cases and the population falling in different compartments of the model. We thus hypothesize that calculating the ratio of unreported deaths to total deaths is a good indicator of inefficiency. Alternatively if the compartments D and Dq are replaced by H and Hq for hospitalisation of hitherto undetected infection and hospitalization of a known infection, the inefficiency can also be interpreted as ratio of 'previously undetected infections at hospitalization' to total hospitalizations.

This paper uses a measure $D_{ratio}$ to describe the ratio of unreported deaths to the total number of deaths obtained from the deceased compartment.

$$D_{ratio} = \frac{D_{undetected}}{D_{detected} + D_{undetected}} \qquad (2)$$

While the structure of the compartment model itself guarantees that the $D_{ratio}$ will asymptotically converge to the inefficiency when parameters of the model are constant, we wish to explore the utility of $D_{ratio}$ in scenarios where inefficiency is varying dynamically. In order to test our hypothesis, we probe the relationship between the $D_{ratio}$ and the intervention inefficiency and their growth trends in a simulated epidemic with changing intervention inefficiencies. In particular we are interested in exploring the scenarios where a drop in efficiencies due to systemic fatigue in a prolonged epidemic can exacerbate a wave in infections.

### Convergence of $D_{ratio}$ to true inefficiency and lag in convergence

The experiments are simulated by seeding the model with an initial set of parameters as described in the Table 1. These parameters are obtained from the understanding of the

**Table 1. The model parameters used for simulations.**

| Parameter | Value |
|---|---|
| $inf_t$—mean infection time | 9 |
| $lat_t$—mean latency time | 7 |
| $in_{phi}$- rate of population inflow | 5 |
| pI—probability of infection in the population inflow | 0.1 |
| Number of days predicted by the simulator | 125 |
| Number of days simulated before the 1st infection | 50 |
| Population size | 10 000 000 |
| a—fraction of infections through contact | 0.3 |
| $\beta$ | 0.35 |

COVID-19 disease as on April 27th 2020 such that the set of parameters such as mean infection time and mean latency time seem epidemiologically plausible. Population is set to 10 million to keep it realistic to a city in India. This is done to ensure the model simulates a realistic number of cases in each of the compartments and it also furnishes an identical backdrop for all the experiments. We can largely classify the parameters as static parameters that don't change in the course of epidemic simulation and dynamic parameters that are time varying and are manipulated throughout the simulation to fit the model to some particular data. Sensitivity analysis has been performed on these static parameters namely The mean infection time ($inf_t$), the mean latency time ($lat_t$), population, and fraction of infections through direct contact (a). Beta and influx are time varying parameters that were modified during the course of the epidemic to simulate different scenarios in the experiment.

Interventions are introduced at two instances of time t1 and t2 having corresponding intervention inefficiencies of Inefficiency1 and Inefficiency2. Each of these Inefficiencies are varied from 0 to 1 in steps of 0.05, and their corresponding time series of $D_{ratios}$ are plotted, thereby exploring the ability of a variety of change patterns.

If the interventions are ineffective, the $D_{ratios}$ increase and similarly when the interventions contain the spread of infection more effectively the $D_{ratios}$ tend to decrease. Hence the Intervention Inefficiency becomes the true value of the $D_{ratios}$. The lag with which the $D_{ratios}$ converge with the prevalent inefficiency value is a good indicator of the utility of this metric. Hence the experiments run with various intervention inefficiencies are noted for observations. The time to convergence is defined as

$$Convergence\ time = T_{\text{latest intervention}} - T_{\text{convergence}} \tag{3}$$

where,

$T_{\text{latest intervention}}$ represents the the day on which the latest intervention was made,

$T_{\text{convergence}}$ is the first day in a block of five days where the $D_{ratio}$ is within ±5% of true inefficiency.

## The effect of window length on $D_{ratio}$ curves

The $D_{ratios}$ can be calculated in multiple ways; by summing over the entire epidemic period, or over windows of several days. The experiment was run over different window periods where the $D_{ratios}$ were smoothed over the window length to omit the baggage of deaths that occurred

much earlier in time. This led to the term $D_{\text{smooth ratio}}$ defined as:

$$D_{\text{smooth ratio}} = D_{\text{ratio}_i} - D_{\text{ratio}_{i-n}} \tag{4}$$

where,

i represents the 'i'th day,

n represents the number of days chosen in the window period.

## Comparison of using $D_{\text{ratios}}$ and daily active cases for corrective action

Laxity in contact tracing and testing are common during a long ongoing epidemic due to systemic fatigue or ignored clusters. This leads to increase in inefficiency and case numbers are subdued or increase only mildly when the epidemic starts to rise. Thus it blindsides the system to a wave building up. We simulate such a case with inefficiency1 = 0.5 and an inefficiency2 rising to 0.7 which results in a wave. A response to this situation results in a decrease in inefficiency to 0.3. The timing of the response and its effect on the impending wave are explored. In this scenario we also explore the resultant rise in $D_{\text{ratios}}$ and active cases and their relative utility in predicting and mitigating the impending latent wave.

## Fitting the model to real world data and validating the utility of the $D_{\text{ratio}}$

The model is fit to three Indian states from the period of Jan 1 2020 to June 06 2022. The model fitting is achieved by tuning the free variables: influx of migrant population, probability of infections from migrant population, beta and percentage of contact tracing parameter c to fit the model to the number of confirmed cases, recoveries and deaths. Transitions from Fig 1 validates that as a consequence fitting the model to confirmed cases, the number of undetected cases also match the real data as they together make up the total population. Further the utility of $D_{\text{ratio}}$ is validated by comparing it with daily active cases and daily new cases by performing a rolling average over different window lengths. The days on which a metric can predict the start of an epidemic wave are identified based on when a metric after being averaged over a rolling window is found to be its highest in the past 20 days. This is used to compare the performance of $D_{\text{ratio}}$ with daily new cases and active cases.

## Results

### The ratio of unreported deaths to total deaths converge with their true value of inefficiency

It may be observed from Fig 2 that the $D_{\text{ratio}}$ tracks the true value of inefficiency as the inefficiency changes dynamically. The metric tracks the true inefficiency on both the increasing and decreasing excursions. While the time to converge varies, the changes in $D_{\text{ratio}}$ follow a largely exponential trajectory. As a result, changes in the metric of interest are large at the onset of change and becomes progressively slower. This suggests that the $D_{\text{ratio}}$ can be a good metric for detecting sudden changes in underlying inefficiency.

### $D_{\text{ratio}}$ metric with varying window sizes and their tracking performance

It may be seen that the $D_{\text{ratio}}$ calculated over the entire epidemic period suffers from memory effects and is unable to converge properly to the true inefficiency. Computing $D_{\text{ratio}}$ over smaller windows results in faster convergence to the true inefficiency. The smaller windows also show larger gradients early on, hence are probably more useful in detecting a change in underlying inefficiency. However this involves some trade-off as with smaller time windows, the metric can become noisy due to daily variations, especially when the absolute number of

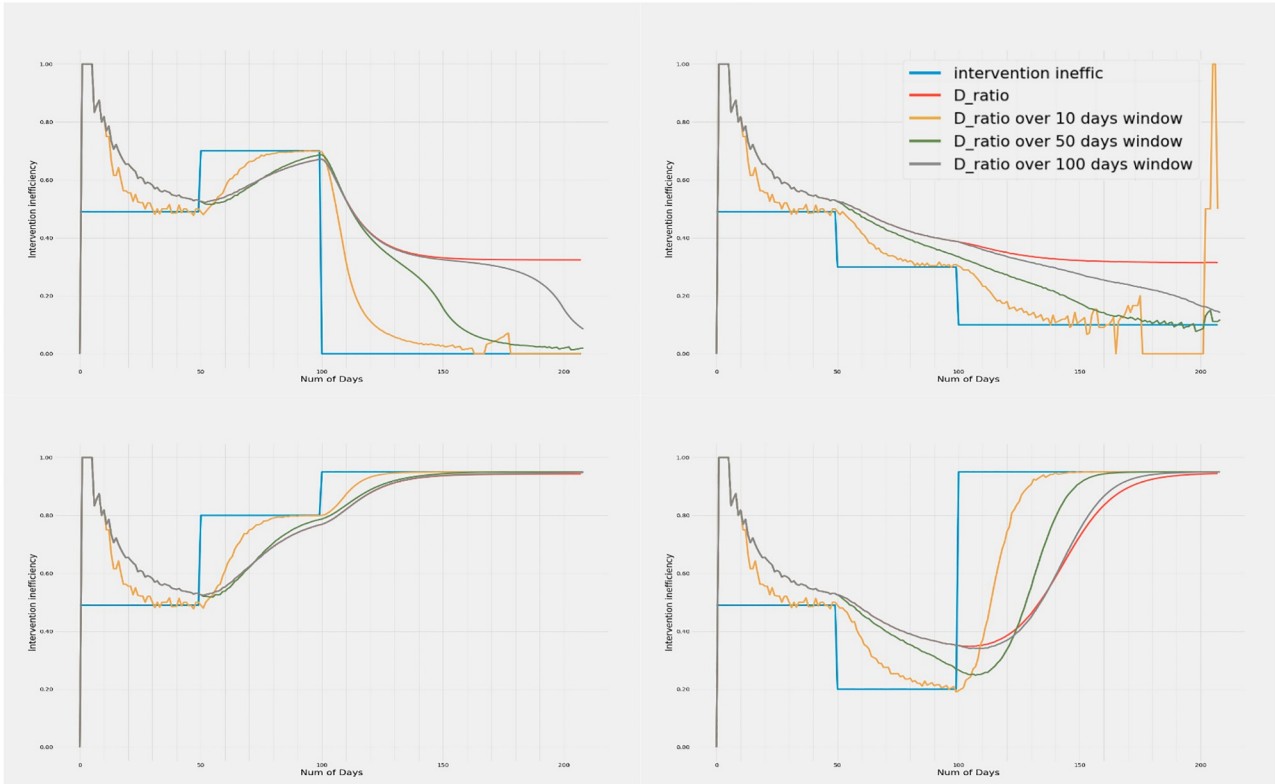

**Fig 2. Variation of intervention inefficiency and $D_{smooth\ ratio}$ with time.** The 4 plots show the trends in $D_{ratio}$ over window periods of 10, 50 and 100 days when the inefficiencies are increased and decreased in different temporal orders. Metrics calculated with all window sizes show high day to day variation when the death numbers in either arm is very low. However, metrics with smaller window sizes are more prone to this daily variation. It is observed that $D_{ratio}$ when smoothed over a small window of 10 days, converges faster but has a lot of variance due to daily fluctuations in numbers. Whereas $D_{ratio}$ over a 100 days window length is smoother, but takes longer to make inferences due to slower convergence. Although sensitive to noise, 10 day window period gives the fastest convergence followed by 50 and 100 day periods.

deaths is very low due to very low inefficiency. The larger windows give smoother convergence and better estimates of underlying inefficiency when the number of absolute deaths in either arm is low. An appropriate window size (or an ensemble thereof) helps achieve a balance between reliability of inefficiency estimates with a faster response time in changing and reforming administrative policies.

## Convergence time to true inefficiency is directly proportional to differences in inefficiencies

The heatmap in Fig 3 shows the relationship between magnitude of change in inefficiency and the number of days for $D_{smooth\ ratio}$ over a 50 day window to converge with the latest inefficiency. The bright diagonal band shows that convergence is the fastest when the difference between the inefficiencies are small and increase with difference between inefficiencies. This diagonal is flanked on both sides by a series of symmetric bands that progressively widen with increase in the inefficiencies. The broader bands of fast convergence in the bottom-right indicates that the metric performs progressively better and is advantageous in higher inefficiencies when the situation most warrants it. The convergences are slower when absolute number of deaths are low in either arm (detected or undetected) as seen when either of the inefficiencies

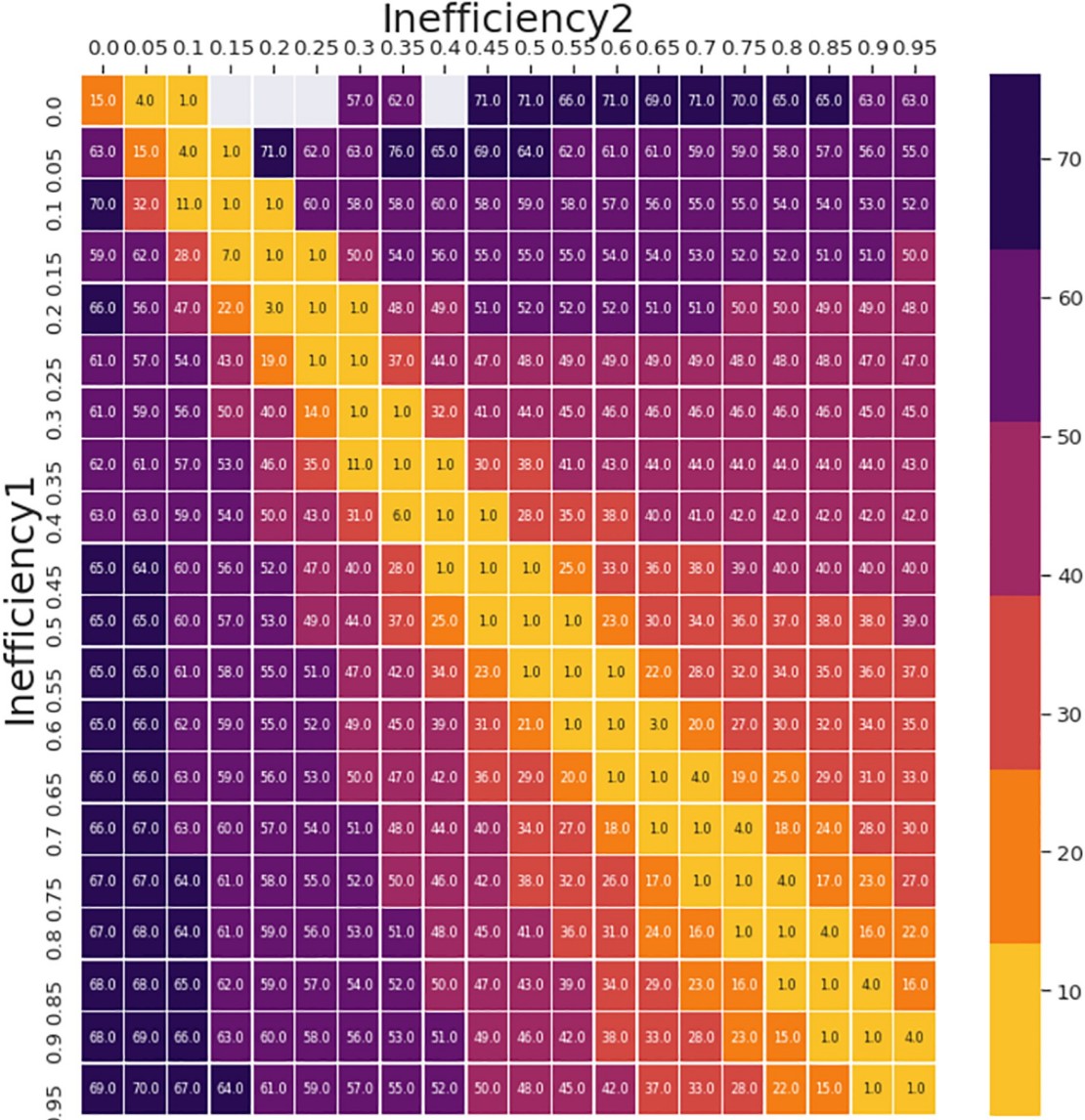

**Fig 3. Correlation between intervention Inefficiency$_1$, Inefficiency$_2$ when D$_{smooth\ ratio}$ is calculated over a 50 day window and the number of days for D$_{smooth\ ratio}$ to converge with Inefficiency$_2$ is plotted on a heatmap.** The days to converge is the least on the diagonal and is largely symmetric. Transitions in the range of inefficiencies 0.2 through 1 converge within a period of 60 days and the range from 0.4 through 1 within 50 days. The empty(grey) cells indicate it takes more number of days to converge than the period of simulation or the cases are extremely low to show up in the D$_{smooth\ ratio}$. In general it may be seen that in the high inefficiency range where it is critical to react fast, the convergences are faster, thus emphasizing the utility of this metric.

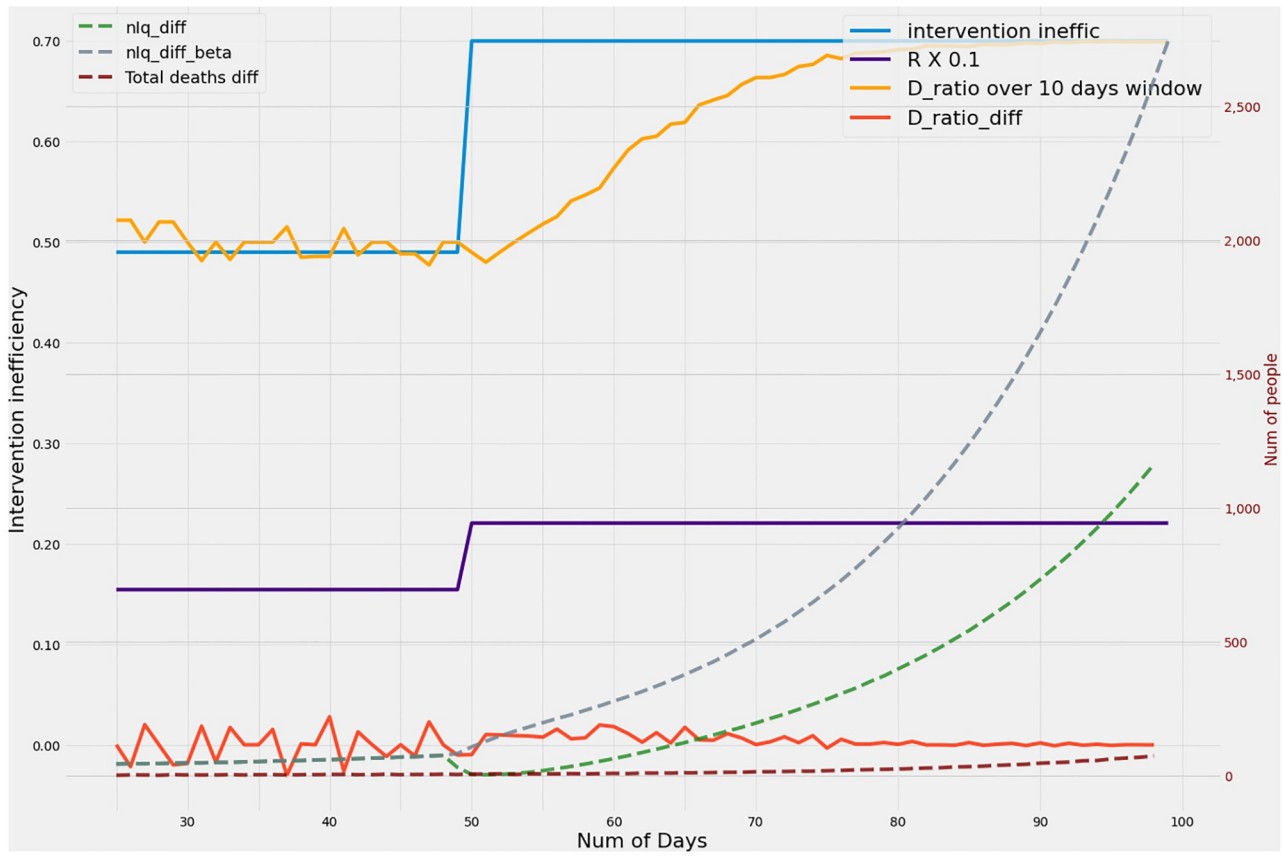

**Fig 4. The graph considers the rise in active cases due to two scenarios, both of which result in identical changes in reproduction number R.** When the rise in R is due to an increase in $\beta$, the active case increase yields a sharper curve giving a true indication of underlying cases. However, when the the increase in R is due to increase in test-track-trace inefficiency, the rise in active cases are slower and in fact even decrease over the short term as the increasing cases are missed due to high inefficiency. A noticeable increase in active cases can only be seen past 30–40 days after the change. However, the increase in $D_{ratio}$ is already evident in 10 days from the day inefficiency increases and stabilizes within a period of 20 days. Complete convergence as defined in Fig 4 comes around 35 days. It may thus be seen that the maximum convergence times of $D_{ratio}$ metrics are comparable to the early rises in active case load based metrics. Thus windowed $D_{ratio}$ based metrics are ideally suited as triggers for initiating mitigation.

is small. Such situations result in high variance of $D_{smooth\ ratios}$ beyond the 5% margins used to define convergence in our definition.

## Smoothed $D_{ratio}$ can detect onset of laxity earlier than other metrics

Fig 4 compares the responses of metrics such as daily increase in active cases and daily deaths with the $D_{ratio}$ calculated over a ten day window. It may be noticed that the change in inefficiency causes an increase in R value from 1.57 to 2.20 thereby precipitating a wave. However responding in the wake of conventional metrics like daily increase in active cases or log changes in active cases are only visible after a time lag as the lower efficiencies result in a large fraction of cases to be undetected. If the same change in $R_t$ had occurred by means of increased $\beta$ instead of an increased inefficiency, the daily increase in active cases would have increased faster, with a possibility of early detection. When inefficiencies increase due to decreased testing or contact tracing, the daily increase in active cases is rather subdued in the short term, masking the impending wave. Although deaths cannot go unnoticed, rise in deaths can take even longer to show up sufficiently to raise an alarm. However, the $D_{ratio}$ captures the

imbalance between the detected and undetected arms and hence indicates the changes underneath. The daily change in the $D_{ratio}$ can be an indicator of the noise in the estimation of $D_{ratio}$. While full convergence of $D_{ratio}$ to the true inefficiency may take time, it is seen from Fig 4 that within a span of about 20 days it converges to the new inefficiency and at 10 days already shows clear indications of rise in inefficiency as evidenced by sudden high increase and moderate variance.

### Mitigation by tracking the $D_{ratio}$ reduces or truncates the epidemic wave intensity or delays it sufficiently

As the number of daily active cases in a pandemic remains low for a long period of time, it is natural for some laxity to creep in the regulations and protocols followed. This results in an increase in the inefficiency as indicated in Fig 5(a) thereby precipitating or advancing an impending latent wave. Intervention resulting in decreased inefficiency can be triggered by one or more indicators. The interventions triggered after observing the conventional metrics like significant rise in the number of daily active cases would easily cost 60 days of valuable time. The earliest response monitoring the situation aggressively would also take no less than 40 days (Fig 5(b) and 5(c)). Despite reducing the deaths and active cases, both these scenarios [Fig 5(b) and 5(c)] leave very little time to prepare or react resulting in quite a number of

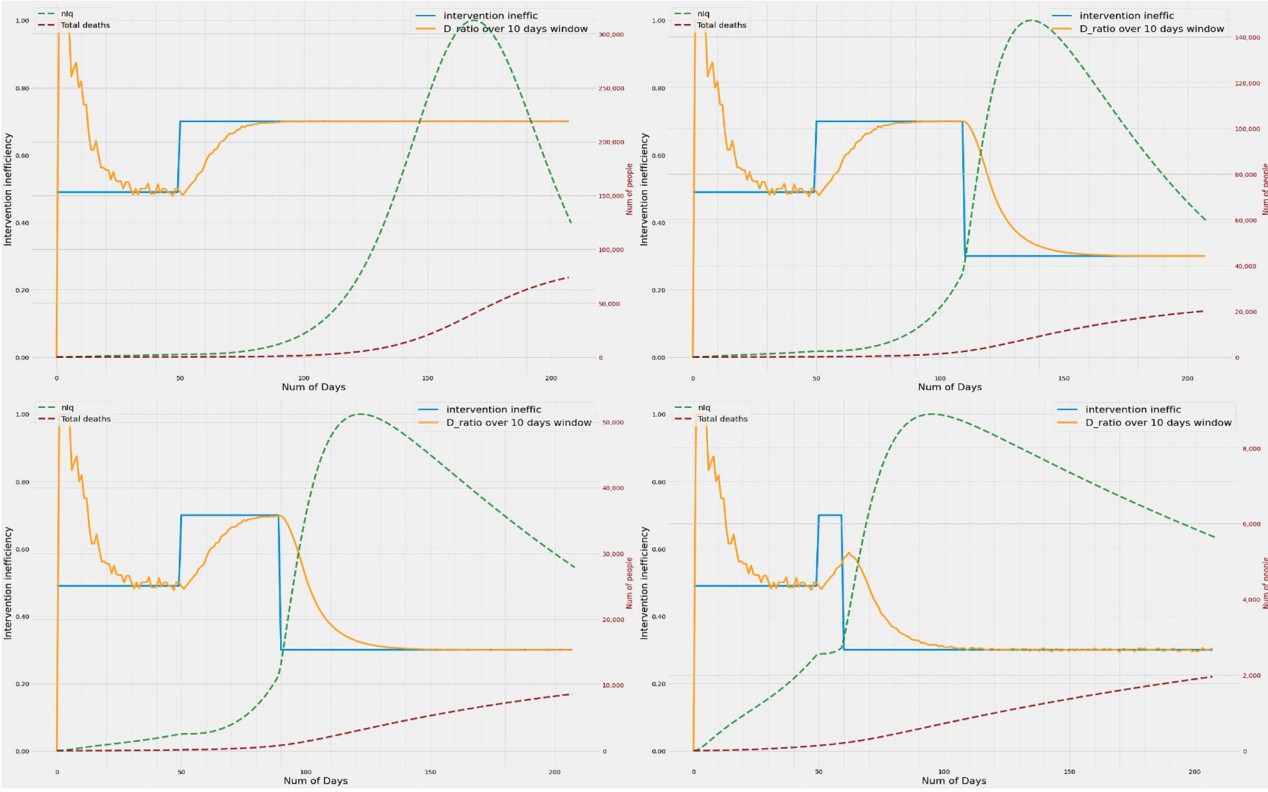

**Fig 5.** Left to right: **(a)** Inefficiency increases, but no counter measures are taken. Result is a peak active case load of 300 000 in a 10 million population and causing close to 75 000 deaths. **(b)** Inefficiency is lowered when the daily active cases significantly increase above 40,000. This measure reduces the daily active cases by nearly 50% and deaths by 75%. **(c)** By monitoring rise in daily active cases closely, lowering the inefficiency when they rise above a smaller threshold of 15 000 active cases daily, the peak and deaths both reduce considerably. **(d)** Instead when the inefficiency is lowered as the $D_{ratio}$ begins to increase, the peak active cases and deaths are averted.

infections and deaths. Intervention based on sudden deviations of the metric $D_{ratio}$ plays a seminal part here in averting the wave or pushing the wave much farther ahead in time as indicated in Fig 5(d), where the inefficiency is lowered based on the increase in $D_{ratio}$. As seen in Fig 4, $D_{ratio}$ can detect the true inefficiency in about 20–30 days with certainty as witnessed by very low daily variations in $D_{ratio}$. The metric can clearly forewarn although with with slightly lower certainty (moderate daily variation) within 10 days. Using $D_{ratio}$ the maximum time to track the inefficiency transition is given by the heatmap in Fig 3 and the earliest response time is about a half or third of it as observed in Fig 4. The benefits of reacting based on this metric is demonstrated in Fig 5(d). The peak active case load and death are reduced significantly. The early warning also provides the administrative bodies sufficient time to plan the interventions, resources and logistics needed to mitigate or tide over it.

## Performance of $D_{ratio}$ and other metrics on real world data

The performance of $D_{ratio}$ is compared with other metrics like daily active cases and daily new cases on a real world epidemic to demonstrate the practical application of the metric. It is observed that a wave is precipitated when beta and influx of migrant population increases and c decreases significantly. The $D_{ratio}$, daily active cases and daily new cases are all compared by taking a n-day rolling average for different values of n (n = 7,10,15,25,50). As demonstrated in Figs 6–8 the metric $D_{ratio}$ always lags the daily active cases and daily new cases when averaged over a 25 day rolling window thereby validating its utility in the purpose of planning mitigation strategies. These results have been validated for rolling windows of different lengths as well. These results prove the advantages of a metric like $D_{ratio}$ in real world scenarios giving the administrative bodies a minimum of 10–15 days to prepare for an upcoming wave and also to bring in interventions by keeping a tab on the underlying model parameters.

## Sensitivity of $D_{ratio}$ to model parameters

The sensitivity analysis for $D_{ratio}$ is performed for the static model parameters that don't vary during the epidemic. The performance of $D_{ratio}$ is tested by varying model parameters like mean infection time (the duration of infection in an individual) and the mean latency time (the period between an individual contracting the infection and when one spreads it to others). The days when $D_{ratio}$, active cases and daily new cases are higher than their respective maximum value in the last 20 days is used to trigger an alert predicting an upcoming wave. The number of days by which $D_{ratio}$ predicts the epidemic wave earlier than active cases and daily new cases are tabulated in the Tables 2 and 3. Although it is found that $D_{ratio}$ is sensitive to these model parameters, it is seen that it is still a better indicator as compared to active cases and daily new cases thereby validating its utility.

## Discussions

Constantly monitoring an epidemic is extremely crucial as even small laxity in test-track-trace efficiencies has great potential to precipitate a wave. This situation can be analysed with the help of parameters like reproductive number, daily increase in daily active cases, test positive ratios etc. But these parameters are largely dependant on testing strategies and the nature of sampling, whereas monitoring deaths are unmissable events in an epidemic and are capable of explaining the underlying situation quantitatively. This paper proposes a metric based on death reports(or equivalently hospitalizations reported) to monitor the inefficiency in test-track-trace performance. We show that $D_{ratio}$ being deaths resulting from hitherto undetected infections as a fraction of all deaths is sensitive to the changes in the inefficiencies prevalent in the system and reflects these changes quicker than any other conventional indicators. Our

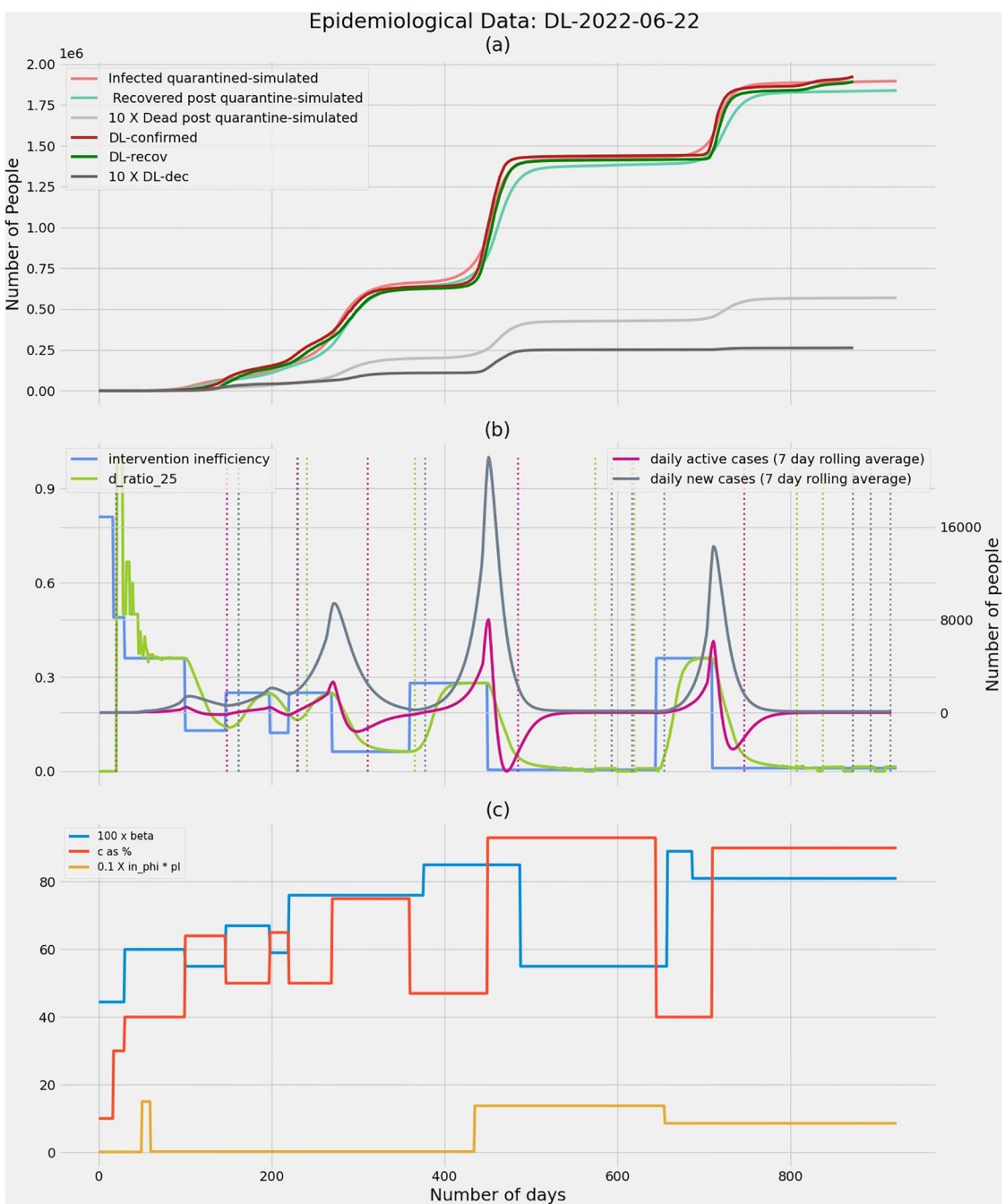

**Fig 6. Delhi as on 22nd June 2022. (a)** Real COVID-19 data and simulated data showing confirmed cases, recoveries and deaths. It is observed that simulations match the real case growth to a great extent. **(b)** The intervention inefficiency underlying the epidemic is plotted along with the ratio of deaths from undetected infections to total number of deaths ($D_{ratio}$), daily new cases and active cases. The dashed vertical lines correspond to the potential time points when an intervention can be made based on the respectively coloured $D_{ratio}$, daily active cases and daily new cases. $D_{ratio}$ predicts an upcoming wave ahead of other metrics when the wave is precipitated due to a drop in efficiency (decreasing values of 'c'—the fraction of infections detected through contact tracing). **(c)** The model parameters beta, fraction of infections that get detected through contact tracing—'c', the fraction of infections spread through migrant population—'in$_{phi}$*pI' are tuned to fit the model to real data.

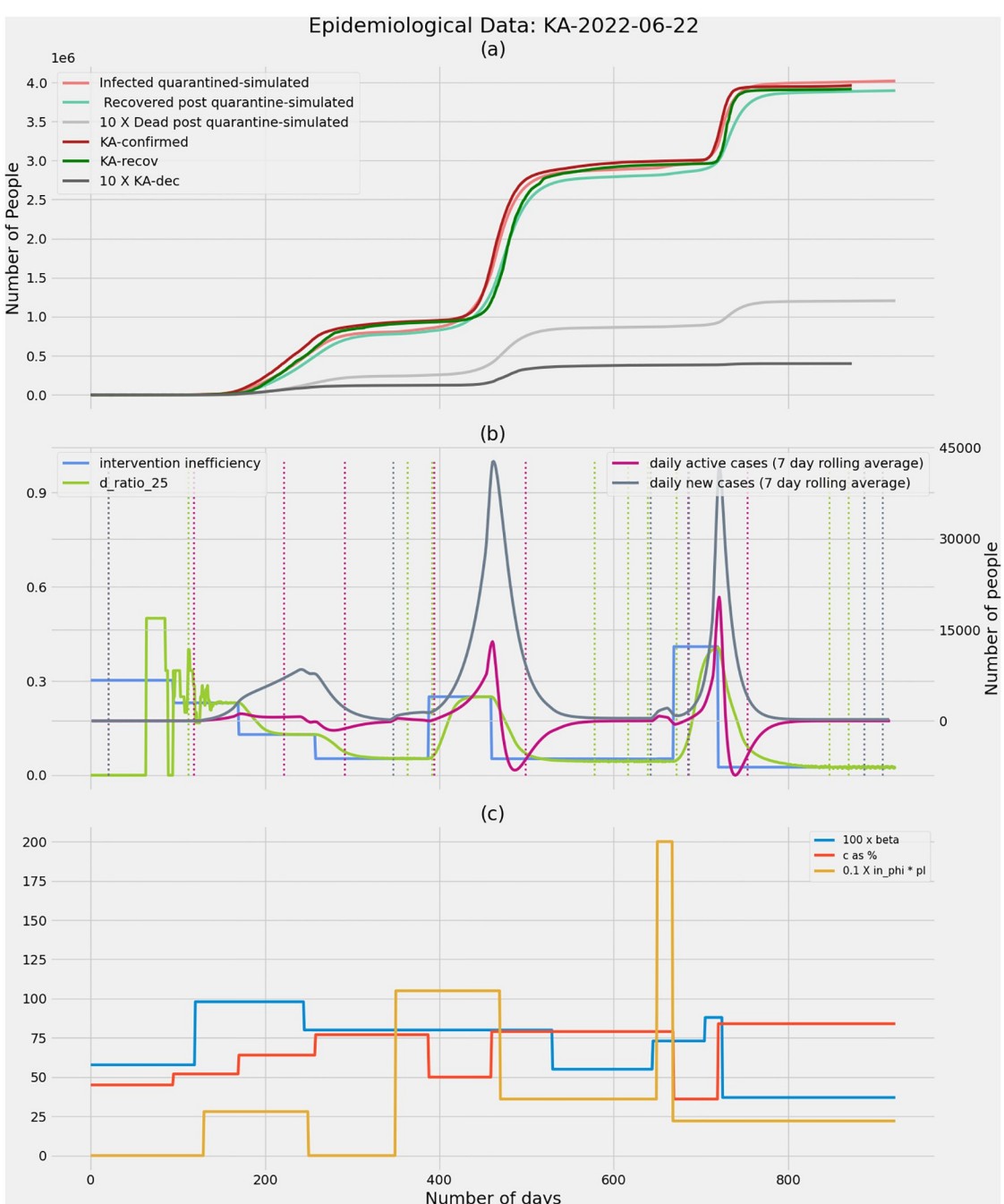

**Fig 7. Karnataka as on 22nd June 2022. (a)** Real COVID-19 data and simulated data showing confirmed cases, recoveries and deaths. It is observed that simulations match the real case growth to a great extent. **(b)** The intervention inefficiency underlying the epidemic is plotted along with the ratio of deaths from undetected infections to total number of deaths ($D_{ratio}$), daily new cases and active cases. The dashed vertical lines correspond to the potential time points when an intervention can be made based on the respectively coloured $D_{ratio}$, daily active cases and daily new cases. $D_{ratio}$ predicts an upcoming wave ahead of other metrics when the wave is precipitated due to a drop in efficiency (decreasing values of 'c'—the fraction of infections detected through contact tracing). **(c)** The model parameters beta, fraction of infections that get detected through contact tracing—'c', the fraction of infections spread through migrant population—'in$_{phi}$*pI' are tuned to fit the model to real data.

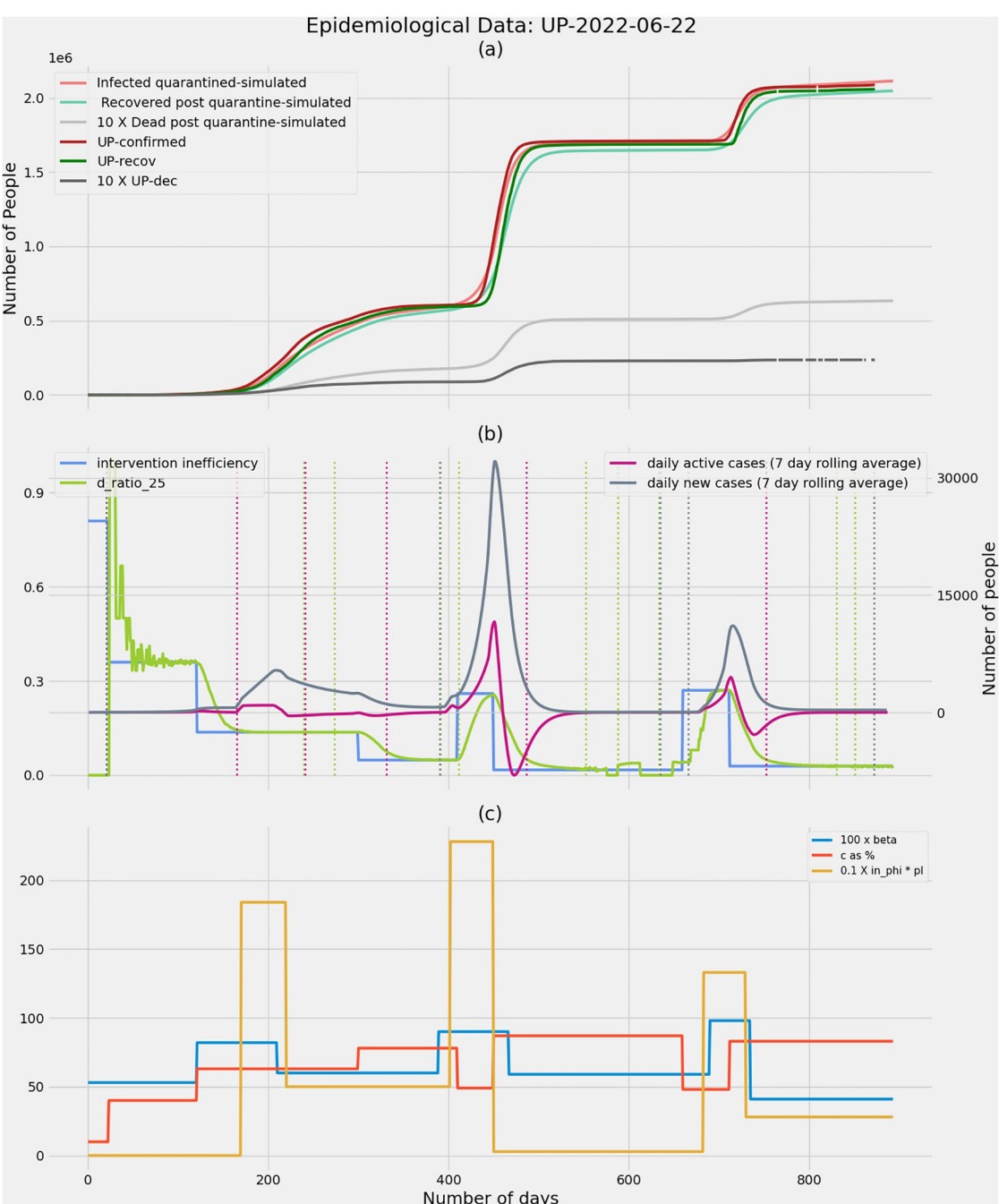

**Fig 8. Uttar Pradesh as on 22nd June 2022. (a)** Real COVID-19 data and simulated data showing confirmed cases, recoveries and deaths. It is observed that simulations match the real case growth to a great extent. **(b)** The intervention inefficiency underlying the epidemic is plotted along with the ratio of deaths from undetected infections to total number of deaths ($D_{ratio}$), daily new cases and active cases. The dashed vertical lines correspond to the potential time points when an intervention can be made based on the respectively coloured $D_{ratio}$, daily active cases and daily new cases. $D_{ratio}$ predicts an upcoming wave ahead of other metrics when the wave is precipitated due to a drop in efficiency (decreasing values of 'c'—the fraction of infections detected through contact tracing). **(c)** The model parameters beta, fraction of infections that get detected through contact tracing—'c', the fraction of infections spread through migrant population—'in$_{phi}$*pI' are tuned to fit the model to real data.

**Table 2. Time lag between $D_{ratio}$ and number of daily active cases (Days).**

| Mean latency period | 3 | 5 | 7 | 9 | 15 | 30 |
|---|---|---|---|---|---|---|
| **Mean infection period** | | | | | | |
| **6** | 40 | 39 | 37 | 35 | 30 | 23 |
| **9** | 70 | 75 | 39 | 37 | 36 | 26 |
| **15** | 50 | 70 | 77 | 83 | 98 | 30 |
| **30** | 45 | 56 | 73 | 79 | 100 | NA |

The table shows the number of days by which $D_{ratio}$ lags(detects earlier) daily active cases in predicting the wave when all other model parameters are kept constant. The columns represent the mean latency period and the rows represent the mean infection period. It is seen that $D_{ratio}$ consistently performs better than daily active cases in the range of latency and infection periods. The cells with NA represent cases when daily active cases failed to predict a wave.

work also highlights the potential reduction in the peak active case load and deaths when mitigation measures are triggered by the metric $D_{ratio}$. The proposed metric and inferences drawn, if employed as part of standard reporting procedures can enable the government to stay better equipped to predict a loss in efficiency and a resulting wave. The benefits can accrue by way of stronger mitigation and prevention of the wave at best or better preparation of hospital infrastructure (beds, ventilators, test kits etc) at the least.

Generalizing this method, the inefficiency thus defined could have been obtained from any of the compartments in the Modified SEIRD model [14] by taking the corresponding numbers from the two parallel arms for reported and unreported cases respectively. As deaths are usually reported at hospitals or government bodies, adequate data can be collected which is not guaranteed in other compartments of the model (namely S,E,I,R). Further these $D_{detected}$ and $D_{undetected}$ can be used interchangeably as $H_{detected}$ and $H_{undetected}$ representing the hospitalized cases or cases requiring intensive care with no loss of generality. $H_{detected}$ would in this case mean hospitalization of a patient already diagnosed with the infection and $H_{undetected}$ would mean the hospitalization of a patient due to severe symptoms but diagnosed with the infection post hospitalization. This extrapolation of results helps in arriving at the analysis faster and more importantly without risking a death.

This model assumes that the rates of death are the same in both the arms—detected and undetected. But in reality it is plausible that the undetected arm could have a lower probability of I→D transition than the detected arm. These different rates would show up as a scaling

**Table 3. Time lag between $D_{ratio}$ and number of daily new cases (Days).**

| Mean latency period | 3 | 5 | 7 | 9 | 15 | 30 |
|---|---|---|---|---|---|---|
| **Mean infection period** | | | | | | |
| **6** | 39 | 38 | 36 | 34 | 30 | 23 |
| **9** | NA | 40 | 38 | 36 | 33 | 26 |
| **15** | NA | NA | NA | 39 | 35 | 28 |
| **30** | NA | NA | NA | NA | NA | 30 |

The table shows the number of days by which $D_{ratio}$ lags(detects earlier) daily new cases in predicting the wave when all other model parameters are kept constant. The columns represent the mean latency period and the rows represent the mean infection period. It is seen that $D_{ratio}$ consistently performs better than daily new cases in the range of latency and infection periods. The cells with NA represent cases when daily new cases failed to predict a wave.

factor of $D_{ratio}$. However, even in such scenarios the gradient of the rise in $D_{ratio}$ compared to the previous epoch would still give indication of the impending wave earlier than the other indicators used currently.

The results of this research indicate that $D_{ratio}$ outperforms the active cases and daily new cases in predicting an upcoming epidemic wave when the epidemic is characterized by a low 'c' typically a scenario as is prevalent currently wherein the amount of testing has come down drastically and contact tracing has become virtually nil. Under these circumstances if an epidemic were to start due to potent imported contagious strain or by any other means, the results here show that it will be quite a while before the epidemic is actually caught using 7 day rolling average of daily new cases or active cases. On the other hand, just tracking the proposed metric $D_{ratio}$ having a very low cost of compliance is a lot more feasible and can catch an upcoming epidemic wave early. Since our metric is specially sensitive at low values of 'c'(infections caught from contact tracing) and 'q'(infections caught from random testing), it is an ideal indicator of an epidemic wave because it imposes least socio economic cost but keeps the surveillance robust. For example, one could put a threshold on the $D_{ratio}$ and escalate within the administration when it hits the threshold. The main advantage of this metric is that it does not force people to get tested, does not impose unnecessary lockdowns, does not disrupt the economics of the state or the country. Hence this scheme is better than tracking 7 day rolling averages of daily new cases or active cases in predicting an epidemic wave in scenarios described above. However when the epidemic wave is precipitated due to changes in beta and influx of population active cases and daily new cases are found to be better indicators of the impending wave. This is a limitation with regard to the utility of the metric.

The model assumes that a person once infected cannot be re-infected. This model doesn't account for changes immunological changes, transmissibility changes due to reinfections, vaccination and mutations of viral strains. These are some limitations with regard to the design of the model.

Further, this model also assumes that the mortality rate is constant across waves during the epidemic and across various strains of viruses. But it has been noticed in reality that there can be different mortality rates in different waves, for different viral strains and also different in vaccinated set of population versus the ones without vaccination. This effect can be observed in the difference in the number of cases simulated in the third wave compared to actual data in Fig (6). It is evident from Fig (7) that the utility of $D_{ratio}$ is still valid.

It has also been observed that the rate of transmission in each limb of the modified SEIRD model can be different based on the age groups, whether symptomatic or asymptomatic, vaccinated or not etc. Hence by incorporating more such endogenous viral parameters in the model it is possible to improve the performance and accuracy of the model.

This model falls short in explaining the changing dynamics in different geographical clusters within a state. Hence it is not possible to infer the localised cause for the surge in cases within a state. There could be a few districts with a denser spread of cases while a slightly farther set of districts might still be unaffected with only fewer cases. [23] talks about the pandemic as a chaotic dynamical system where the growth of the pandemic is unpredictable due to a vast set of changing parameters across countries. But this unpredictability affects all the metrics like daily active cases, daily new cases, and $D_{ratio}$ equally as they are all inferred from the same field data of a specific region. Since the purpose of this metric is to indicate an impending wave in a region, the utility of the metric holds good regardless of the chaotic nature of the epidemic.

Ideally a model with all the above described parameters will provide a more comprehensive picture of the ongoing epidemic. But an epidemic due to a completely new pathogen can go full blown even before most of these variables are understood. We demonstrate that a model

with a minimal set of parameters can still capture the $D_{ratio}$ earlier than other conventional metrics. The advantage of such a model comes to light when non experts, people without domain knowledge can use the model to make timely interventions.

In light of the above results, we strongly propose that policy makers and healthcare administrators consider the inclusion of $D_{ratio}$ metric as part of their decision making framework. Since it is likely that hospitals already have this data, implementing this proposal would only demand certain changes in the book keeping of already available data, posing minimal overheads. By closely monitoring the trends in this metric it is possible to detect the changes in the laxity in regulations and take corrective measures much earlier and also gain ample time to work on the strategies and procurement of required resources to overcome the wave.

## Author Contributions

**Conceptualization:** Yashaswini Mandayam Rangayyan, Mohan Raghavan.

**Data curation:** Yashaswini Mandayam Rangayyan, Sriram Kidambi.

**Formal analysis:** Yashaswini Mandayam Rangayyan, Mohan Raghavan.

**Investigation:** Mohan Raghavan.

**Methodology:** Yashaswini Mandayam Rangayyan.

**Project administration:** Mohan Raghavan.

**Software:** Yashaswini Mandayam Rangayyan, Sriram Kidambi.

**Supervision:** Mohan Raghavan.

**Writing – original draft:** Yashaswini Mandayam Rangayyan.

**Writing – review & editing:** Mohan Raghavan.

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
