## [Decision Letter · Decision Letter 0]

17 May 2022

PONE-D-22-02784Deaths from hitherto undetected Covid-19 infections as a fraction of Covid-19 deaths can be an indicator of the extent of latent infection in populationsPLOS ONE

Dear Dr. M R,

Thank you for submitting your manuscript to PLOS ONE. After careful consideration, we feel that it has merit but does not fully meet PLOS ONE’s publication criteria as it currently stands. Therefore, we invite you to submit a revised version of the manuscript that addresses the points raised during the review process.

We look forward to receiving your revised manuscript.

Kind regards,

Alessandro Borri

Academic Editor

PLOS ONE

Journal Requirements:

Reviewers' comments:

Reviewer's Responses to Questions

**Comments to the Author**

1. Is the manuscript technically sound, and do the data support the conclusions?

Reviewer #1: Partly

Reviewer #2: Yes

2. Has the statistical analysis been performed appropriately and rigorously? 

Reviewer #1: I Don't Know

Reviewer #2: Yes

3. Have the authors made all data underlying the findings in their manuscript fully available?

Reviewer #1: No

Reviewer #2: Yes

4. Is the manuscript presented in an intelligible fashion and written in standard English?

Reviewer #1: Yes

Reviewer #2: Yes

5. Review Comments to the Author

Reviewer #1: Thank you for providing me with an opportunity to review this article. I appreciate the authors for working on an important topic. The authors have proposed an indicator as a measure to estimate the extent of latent infection in populations. There is some ambiguity with the study.

for example, although the authors have targeted the COVID-19 pandemic, in the method section the parameters are presented generally and not specified to COVID-19. Moreover, the authors' interpretation seems as if the formulas and parameters used in the model can be used in any pandemic, while every pandemic might have a different pattern undermining the proposed parameters.

It seems that there is a difference between the modeling estimates and real-world results in the study. I kindly recommend the authors compare the real data (COVID-19 data) with simulation results in their modeling study to investigate the accuracy of the simulation. For example, in their figures, the authors have estimated the pandemic with a single peak in 200 days; whereas in the real world, we have seen several peaks within such period. And each peak has different parameter quantities.

Also, it is not clear how the authors have calculated the number of undetected cases (D) and what is their rational to do so in different types of pandemics.

More importantly, we use any indicator for a specific purpose. However, in the current study, it is not clear for what specific purpose the proposed indicator should be used. Comparing the different indicators depends on the purpose.

I also suggest the authors strengthening their paper by using a few more references and discussing their results comparing them with other studies in similar contexts.

Reviewer #2: [1] This article uses a modified SEIRD model to show that the ratio of deaths and hospitalizations from an undetected infection to total deaths converges to a measure of systemic test-trace inefficiency. I think there is enough new content in this paper to distinguish it from other works, considering that in recent years many articles have been published on the use of mathematical models for the dynamics of the COVID-19 pandemic, but without any practical utility. However, before proposing that policy makers and health administrators should consider including the Dratio metric as part of their decision-making framework, it is necessary that this paper is not merely a mathematical application of a model but also considers important biological assumptions. In a recent article, Jones and Strigul (2021) hypothesized that the unpredictability of the COVID-19 pandemic could be a fundamental property if the disease spread is a chaotic dynamical system. This means that the change in daily numbers of COVID-19 is affected by a very large number of factors, such as the population's adherence to prevention measures, vaccination, social isolation, and new variants of the virus. In addition, in a recent letter published in the Journal of Medical Virology, Divino and colleagues advert that models used to generate predictions, scenarios, and projections about COVID-19 infections and hospitalizations, are unreliable (https://doi.org/10.1002/jmv.27325). Therefore, the authors must describe the scope and potential limitations of the proposed model from a clinical and epidemiological, not only mathematical, perspective, justifying how these exogenous variables can affect the results obtained. Future perspectives for more complex models, including other factors, are also critical.

Reference: Jones A, Strigul N. Is spread of COVID-19 a chaotic epidemic? Chaos Solitons Fractals. 2021;142:110376.

[2] Please justify the parameters shown in Table 1.

[3] Please provide a sensitivity analysis considering changes in the parameters shown in Table 1.

6. PLOS authors have the option to publish the peer review history of their article (what does this mean?). If published, this will include your full peer review and any attached files.

Reviewer #1: No

Reviewer #2: No

---

## [Author Response · Author response to Decision Letter 0]

18 Jul 2022

Thank you very much for your valuable suggestions and constructive critique of our manuscript. We have addressed all the review comments in our "Response to reviews.pdf" that has been submitted along with the manuscript. We have also incorporated these suggestions in our manuscript and have made appropriate changes to it.

---

## [Decision Letter · Decision Letter 1]

6 Sep 2022

PONE-D-22-02784R1Deaths from hitherto undetected Covid-19 infections as a fraction of Covid-19 deaths can be an indicator of the extent of latent infection in populationsPLOS ONE

Dear Dr. M R,

Thank you for submitting your manuscript to PLOS ONE. After careful consideration, we feel that it has merit but does not fully meet PLOS ONE’s publication criteria as it currently stands. Therefore, we invite you to submit a revised version of the manuscript that addresses the points raised during the review process.

We look forward to receiving your revised manuscript.

Kind regards,

Alessandro Borri

Academic Editor

PLOS ONE

Journal Requirements:

Additional Editor Comments:

Note that, based on the current Reviews, future acceptance of the paper is not guaranteed and failure in addressing the Referees' comments may still lead to the rejection of the manuscript.

Reviewers' comments:

Reviewer's Responses to Questions

**Comments to the Author**

1. If the authors have adequately addressed your comments raised in a previous round of review and you feel that this manuscript is now acceptable for publication, you may indicate that here to bypass the “Comments to the Author” section, enter your conflict of interest statement in the “Confidential to Editor” section, and submit your "Accept" recommendation.

Reviewer #1: All comments have been addressed

Reviewer #3: (No Response)

2. Is the manuscript technically sound, and do the data support the conclusions?

Reviewer #1: Yes

Reviewer #3: Partly

3. Has the statistical analysis been performed appropriately and rigorously? 

Reviewer #1: I Don't Know

Reviewer #3: Yes

4. Have the authors made all data underlying the findings in their manuscript fully available?

Reviewer #1: Yes

Reviewer #3: Yes

5. Is the manuscript presented in an intelligible fashion and written in standard English?

Reviewer #1: Yes

Reviewer #3: No

6. Review Comments to the Author

Reviewer #1: (No Response)

Reviewer #3: OVERVIEW

While it is important to examine the potential for new metrics for monitoring epidemics, as the authors suggest here their D_ratio of deaths to hospitalizations, there is still a lack of clarity. The manuscript Introduction should begin describing the COVID-19 pandemic or at least epidemics, and the manuscript definitely needs further proofreading/editing.

COMMENTS ON THE AUTHORS’ RESPONSE TO REVIEWERS

Response to reviewer #1, Q2

In Figure 1, could the authors ensure there are panel labels (a, b, c) rather than top to bottom, the figure caption needs proofreading edits. Perhaps it would be clearer to refer to ‘real’ COVID-19 data as actual data. The legend text in panel (a) is too small to read. Define the ‘D_ratio’ on first use in this figure for clarity. No need to capitalize ‘Active’ cases. No need to include ‘it is evident’. Ensure the legend is panel (b) is fully spelled out, i.e., ‘intervention ineffc’. In the description for panel (c) what do the authors mean by ‘free’ variables? Not fixed? Also define ‘c’ in the figure caption (last character in the response). Between figures 1 and 3 panels (b), can the authors explain why there is a difference in the number of days of lag, i.e., 36 days for Delhi vs 58 days for Karnataka vs 31 days for UP– unless this has been covered in the main manuscript. For Figure 3, the description for panel (c) notes that these free variables were modified to fit the epidemic, but from the figure the fitting is not clear. Please explain or modify. In the response the authors note the model was fit to data from 1 January 2020 to 6 June 2022, but in the figure captions the dates are listed as 22 June 2022, why the difference? Most often new daily cases are reported as a 7-day rolling average, not as the authors note (daily new cases) surrounding the note around noisiness potentially leading to more false positive waves. Again, in the authors’ response this method may raise a warning flag, but the message around needing to bring in interventions needs to be placed in the context of pandemic fatigue.

Response to reviewer #1, Q3

Can the authors be very clear that if they fit to confirmed cases, the undetected cases they estimate would be in addition to the confirmed cases that are fit, for a total estimated cases (confirmed + undetected)?

Response to reviewer #1, Q4

I agree with the reviewer’s statement. The authors claim ranking needs sufficient data, like death rates; however, their ratio metric also relies on reporting of numbers of deaths (which as noted below may be underreported). Would the same issue apply?

Response to reviewer #2, Overall statement

I agree as well with this reviewer, adding social and economic assumptions must also be considered.

Response to reviewer #2, Q7

How do the author select settings where their response statement is true “growth of epidemic in a particular state where there is less variance in the underlying parameters”? What is their measure and threshold for variance? And could they please confirm or list the underlying parameters in question in their response (are they referring to beta, c, q, etc. as noted in the next sentence)? To note, Divino are referring to projections of hospitalizations, but the authors refer to hospitalization data – do they mean projections/estimates or reported hospitalizations (to which they refer to as ‘real’ data)? Please be clear in the response, otherwise, it is confusing to assess the validity of the response.

Response to reviewer #2, Q8

To the authors’ point (1) why is reinfection not modelled? This is not representative of the real-world situation. Is it for simplicity sake? If so, please state this in the response and reflect this in the manuscript if this has not been done. Also, please state this as a limitation if not done, and provide a hypothesis as to how this might affect projections. Not accounting for the effect of vaccinations and viral variant are major limitations. The authors must defend this choice and clearly show the effect of these limiting assumptions if this method is to be published. This should almost be mentioned in the abstract, so readers can assess the gravity.

Response to reviewer #2, Q11

For the table labelled as Figure 4 (it is really a table and should be labelled as such) in the table header row please label the units for the columns (days), and explain the variables in the row headers in the figure (table) caption. Also describe the shading. In the paragraph below the caption, the authors state ‘when the mean infection times are very high’, how high is very high? They note ‘mean’, but this is not specified in the table nor the caption (nor is ‘average’), as well as mean latency times – latency is not shown in the table in Figure 4 – where/how does this come into play here? Please be clear.

ADDITIONAL COMMENTS

MAJOR COMMENTS

1. Can the authors speak to the issue of underreported COVID-19-related deaths early in the manuscript?

2. The Introduction leads with a detailing of compartmental models. This seems to be a lost opportunity, whereby the authors should set the scene for the focus of this study. Extensive detailing of compartmental models should be reserved for the Methods.

MINOR COMMENTS

1. The word choice ‘hitherto’ in the title and abstract and ‘flatter’ in the abstract may not be appropriate choices.

2. In the abstract, is the drop in efficiency specific to any point in the pathway (e.g., successful treatment) or at any point in the pathway?

3. The last sentence of the abstract, ‘Further hospitals may need to be mandated to distinctly register deaths/hospitalizations due to previously undetected infections.’, first, a comma should be inserted after ‘Further’. Second, will it be possible for hospitals to be mandated to register the deaths/hospitalizations ratio? In which setting or settings would this be possible? Third, it is unclear and should be even in the abstract, how any proportion of this ratio is due to undetected infections? If infections were detected, is the point the authors are making is that deaths and hospitalizations could be prevented through ambulatory treatment? This needs to be clarified.

4. From the introduction, this sentence should be reworded, ‘Hence the authors say that models fit to hospitalizations and deaths are more reliable’, particularly the word ‘say’. Also, how are these model more reliable, with respect to what process (calibration) and/or outcome (forecasting for which outcome indicator)? Be more specific.

MINOR FORMATTING

1. The first sentence of the Introduction, spaces are missing after commas ‘SIS,SIR,SEIR’.

2. Make use of ‘covid-19’ and ‘covid19’ consistent throughout the manuscript.

7. PLOS authors have the option to publish the peer review history of their article (what does this mean?). If published, this will include your full peer review and any attached files.

Reviewer #1: No

Reviewer #3: No

---

## [Author Response · Author response to Decision Letter 1]

10 Feb 2023

All the review comments by the reviewers have been answered in the 'response to reviewers.pdf' that has been uploaded and the necessary changes have been incorporated in the manuscript.

---

## [Decision Letter · Decision Letter 2]

2 Mar 2023

Deaths from  undetected COVID-19 infections as a fraction of COVID-19 deaths can be used for early detection of an upcoming epidemic wave

PONE-D-22-02784R2

Dear Dr. M R,

We’re pleased to inform you that your manuscript has been judged scientifically suitable for publication and will be formally accepted for publication once it meets all outstanding technical requirements.

Kind regards,

Alessandro Borri

Academic Editor

PLOS ONE

Additional Editor Comments (optional):

Reviewers' comments:

Reviewer's Responses to Questions

**Comments to the Author**

1. If the authors have adequately addressed your comments raised in a previous round of review and you feel that this manuscript is now acceptable for publication, you may indicate that here to bypass the “Comments to the Author” section, enter your conflict of interest statement in the “Confidential to Editor” section, and submit your "Accept" recommendation.

Reviewer #3: All comments have been addressed

2. Is the manuscript technically sound, and do the data support the conclusions?

Reviewer #3: Yes

3. Has the statistical analysis been performed appropriately and rigorously? 

Reviewer #3: Yes

4. Have the authors made all data underlying the findings in their manuscript fully available?

Reviewer #3: Yes

5. Is the manuscript presented in an intelligible fashion and written in standard English?

Reviewer #3: Yes

6. Review Comments to the Author

Reviewer #3: All previous comments have been addressed. I have no further comments.

7. PLOS authors have the option to publish the peer review history of their article (what does this mean?). If published, this will include your full peer review and any attached files.

Reviewer #3: No

---

## [Editor Report · Acceptance letter]

9 Mar 2023

PONE-D-22-02784R2 

Deaths from undetected COVID-19 infections as a fraction of
COVID-19 deaths can be used for early detection of an
upcoming epidemic wave 

Dear Dr. Mandayam Rangayyan:

I'm pleased to inform you that your manuscript has been deemed suitable for publication in PLOS ONE. Congratulations! Your manuscript is now with our production department. 

Kind regards, 

on behalf of

Dr. Alessandro Borri 

Academic Editor

PLOS ONE